# Geospatial Least Squares Support Vector Regression Fused with Spatial Weight Matrix

**Haiqi Wang** [1], **Liuke Li** [1,*], **Lei Che** [2], **Haoran Kong** [1], **Qiong Wang** [1], **Zhihai Wang** [1] **and Jianbo Xu** [1]

1   College of Oceanography and Space Informatics, China University of Petroleum, Qingdao 266580, China; wanghaiqi@upc.edu.cn (H.W.); z19160014@s.upc.edu.cn (H.K.); 1601050210@s.upc.edu.cn (Q.W.); z20160101@s.upc.edu.cn (Z.W.); z20160105@s.upc.edu.cn (J.X.)
2   China Research Institute of Radio Propagation, Qingdao 266580, China; che1@crirp.ac.cn
*   Correspondence: s19160013@s.upc.edu.cn

**Abstract:** Due to the increasingly complex objects and massive information involved in spatial statistics analysis, least squares support vector regression (LS-SVR) with a good stability and high calculation speed is widely applied in regression problems of geospatial objects. According to Tobler's First Law of Geography, near things are more related than distant things. However, very few studies have focused on the spatial dependence between geospatial objects via SVR. To comprehensively consider the spatial and attribute characteristics of geospatial objects, a geospatial LS-SVR model for geospatial data regression prediction is proposed in this paper. The 0–1 type and numeric-type spatial weight matrices are introduced as dependence measures between geospatial objects and fused into a single regression function of the LS-SVR model. Comparisons of the results obtained with the proposed and conventional models and other traditional models indicate that fusion of the spatial weight matrix can improve the prediction accuracy. The proposed model is more suitable for geospatial data regression prediction and enhances the ability of geospatial phenomena to explain geospatial data.

**Keywords:** spatial weight matrix; spatial prediction; least squares support vector regression (LS-SVR)

## 1. Introduction

Spatial statistics analysis refers to the description and analysis of spatial phenomena from the perspective of geography. As a research object of spatial statistics analysis, spatial lattice data are data retrieved from spatially random processes in which the number of collected sites is countable [1]. It has a geographical location and is a digital description of the spatial and attribute characteristics of geospatial objects. The data may not obey a normal distribution, and there may exist complex nonlinear relationships among the variables.

Due to the complex structure of spatial lattice data, traditional linear statistical methods do not accurately resolve practical geographical problems such as linear regression of geographical data and partial least squares estimators. Therefore, intelligent computing techniques have been gradually developed to address spatial analysis problems. Support vector regression (SVR) is a popular supervised machine learning algorithm based on the support vector machine (SVM). The theoretical foundation of the SVR model is statistical learning theory, in which inference rules not only consider the generalization performance but also pursue an optimal solution given limited samples [2]. The SVR model relies on a nonlinear function to map input data into a high-dimensional feature space, and a linear classification is then computed in the mapped feature space, which facilitates relatively simple mathematical calculations [3] and provides the ability to learn nonlinear relationships between the variables [4]. Due to its good performance in regard to statistical analysis problems with a small sample size, nonlinearity, high dimensions or local optimum values, this method is widely applied in missing data imputation [5,6], random error analysis [7,8],

remote sensing image inversion [9,10] and prediction of traffic flow conditions [11,12], soil properties [13,14], and temperature [15] and other spatial data mining problems [16].

As an improvement of the standard SVM model, the least squares support vector regression (LS-SVR) [17] model substitutes Vapnik's $\varepsilon$-insensitive loss function with a sum-squared error (SSE) cost function, replaces the inequality constraints of the standard SVR model with equality constraints and solves the optimization problem by constructing linear equations. Due to the application of equality constraints, the solution process is transformed from a quadratic programming problem into a system of linear equations obtained by the Karush–Kuhn–Tucker (KKT) condition, which greatly reduces the solution difficulty. The LS-SVR model has been improved for different application fields by dividing spatiotemporal factors into different groups [18], allocating various weights to different data [19] and combining the model with optimization algorithms such as the genetic algorithm (GA) [5,7,20], particle swarm optimization (PSO) [6,21–24] and artificial bee colony (ABC) [25] to achieve automatic optimal parameter selection. This improvement of the LS-SVR model adopts new ways in variable integration or parameter selection, but no improvement is made according to the essential characteristics of spatial lattice data.

Tobler's First Law of Geography states that all attribute values on a geographic surface are related, but closer objects are more strongly related than are more distant ones [26]. Spatial dependence, also referred to as spatial autocorrelation, denotes the variance at a small scale. This suggests that nearby sites tend to possess similar characteristics and thus exhibit spatial continuity. Machine learning methods fused with spatial autocorrelation can improve the model performance [27]. The traditional LS-SVR model does not provide the inherent capability of handling geospatial data. The spatio-temporal SVR model has been invented by some researchers and tried to be used in the prediction of temperature, traffic flow or other geospatial data with temporal information [28–30]. With the addition of spatial information, the model performs better [31]. Therefore, it is necessary to consider spatial information when implementing the LS-SVR model.

The spatial weight matrix is the formal expression of the spatial dependence between observations [32]. The spatial weight matrix can be divided into two categories: the 0–1 type and numeric-type spatial weight matrices. In the 0–1 type matrix, the element value is either 0 or 1 according to a specified measurement criterion, such as the spatial contiguity or spatial threshold distance [33]. The numeric-type spatial weight matrix, in which the element value is a numeric value, can be created based on the spatial distance, shared-boundary length, area of the geographical object and any combination [34,35]. Certain complex spatial weight matrices are constructed based on the geographic statistical information between geospatial objects [36,37] or based on the social and economic distances under a certain problem [38].

In this study, we proposed the geospatial LS-SVR model, which is an LS-SVR model integrating the spatial dependence among geospatial objects, to perform regression prediction of geospatial data. Two different types of spatial weight matrices were fused into the regression function of the LS-SVR model to measure the strength of the interaction between geospatial objects. We considered three datasets to evaluate the accuracy of the proposed geospatial LS-SVR model and conventional LS-SVR model. It was confirmed that fusing considering spatial autocorrelation could improve the prediction accuracy. The proposed model could not only retain the good generalization performance of the SVR model but could also reflect the essential characteristics of geospatial objects.

## 2. Methodology

### 2.1. Problem Description

Since the spatial weight matrix is only applicable to lattice data, we only consider lattice data with a countable site index. Point feature datasets should be converted into polygon feature datasets. Suppose there are N geospatial objects in geographical region S, i.e., $S = \{s_1, s_2, \ldots, s_N\}$. Given a geospatial object $s_i$, its location or center location is $(p_i, q_i)$, and its M-dimensional attribute vector is $Attr(s_i) = [a_{i1}, a_{i2}, \ldots, a_{iM}]$. When a

particular attribute of $s_i$ $(i = 1, 2, \ldots, N)$ depends on its other d (d < M) attributes, the former is referred to as the dependent variable, denoted as $y_i$, and the latter are referred to as the explanatory variables, denoted as $x_i = [a_{ik}, \ldots]$, where $k \in \{1, 2, \ldots, M\}$.

Given an observation dataset $\{(x_i, y_i)\}_{i=1}^{N}$ comprising N geospatial objects $\{s_i\}$, where $x_i \in R^d$ and $y_i \in R$. The general form of regression model is a regression function $y = f(x)$ that reflects the dependency between $y$ and $x$, where $x = [x_1, x_2, \ldots, x_N]^T$ and $y = [y_1, y_2, \ldots, y_N]^T$. This function can be linear function or nonlinear function. In regard to regression modeling of geographical data, according to Tobler's first law of geography, all attribute values on a geographic surface are related. That is, if geospatial object $s_i$ and geospatial object $s_j$ are adjacent or close, the variation in the dependent variable $y_i$ of geospatial object $s_i$ not only depends on the variation in the explanatory variables $x_i$ of $s_i$ but also depends on the variation of geospatial object $s_j$. Therefore, the general geographical regression form based on spatial autocorrelation can be expressed as:

$$y_i \approx f(x_i, x_j, y_j), i = 1, 2, \ldots, N \text{ and } j \in [1, 2, \ldots, N], \tag{1}$$

where $x_i$ and $y_i$ are the attributes of $s_i$, and $x_j$ and $y_j$ are the attributes of $s_j$.

The spatial dependence between any two geospatial objects $s_i$ and $s_j$ can be quantitatively measured by the spatial weight matrix $W_{N \times N}$. Concretely, for $s_i$, the spatial dependence between other objects $s_j$ $(j = 1, 2, \ldots, N \text{ and } j \neq i)$ and $s_i$ is reflected by element $w_{ij}$ of W. The larger the value of $w_{ij}$ is, the stronger the spatial dependence between $s_i$ and $s_j$ is. $w_{ij} = 0$ suggests that there exists no spatial autocorrelation between $s_i$ and $s_j$. Therefore, $w_{ij}$ can be introduced into Equation (1) as a weighting factor, which reflects the magnitude of the influence of $x_j$ of $s_j$ on the dependent variable $y_i$ of $s_i$, and Equation (1) can be further expressed as:

$$y_i \approx f(x_i, w_{ij}x_j, w_{ij}y_j), \tag{2}$$

The geospatial LS-SVR model relies on a similar form of $Wx$ to fuse spatial autocorrelation into the regression function $\omega^T \varphi(x) + b$ of the SVR model, which not only maintains the excellent features of the SVM model but also reflects the spatial dependence of geographical data.

### 2.2. Spatial Weight Matrix

In the study area $S = \{s_i\}_{i=1}^{N}$, the spatial weight matrix W is an N × N matrix, and element $w_{ij}$ $(i = 1, 2, \ldots, N, \text{ for } j = 1, 2, \ldots, N \text{ and } j \neq i)$ expresses and measures the spatial relationship between $s_i$ and $s_j$. $w_{ij} = w_{ji}$ indicates that the spatial relationship from $s_j$ to $s_i$ is the same as the spatial relationship from $s_i$ to $s_j$, and $w_{ij} \neq w_{ji}$ indicates that the spatial relationship between $s_j \to s_i$ and $s_i \to s_j$ differs. The general form of the spatial weight matrix W is:

$$W = \begin{bmatrix} w_{11} & w_{12} & \cdots & w_{1N} \\ w_{21} & w_{22} & \cdots & w_{2N} \\ \vdots & \vdots & \cdots & \vdots \\ w_{N1} & w_{N2} & \cdots & w_{NN} \end{bmatrix}, \tag{3}$$

In this paper, we examined two types of spatial weight matrices, namely, the 0–1 type and numeric-type spatial weight matrices.

Regarding the 0–1 type spatial weight matrix, the first-order queen contiguity matrix was adopted in this paper. This matrix applies both common edges and vertices to define contiguous objects. $w_{ij} = 1$ indicates that there is a common edge or vertex between $s_i$ and $s_j$, while $w_{ij} = 0$ indicates that there is no common edge or vertex between $s_i$ and $s_j$ (Figure 1).

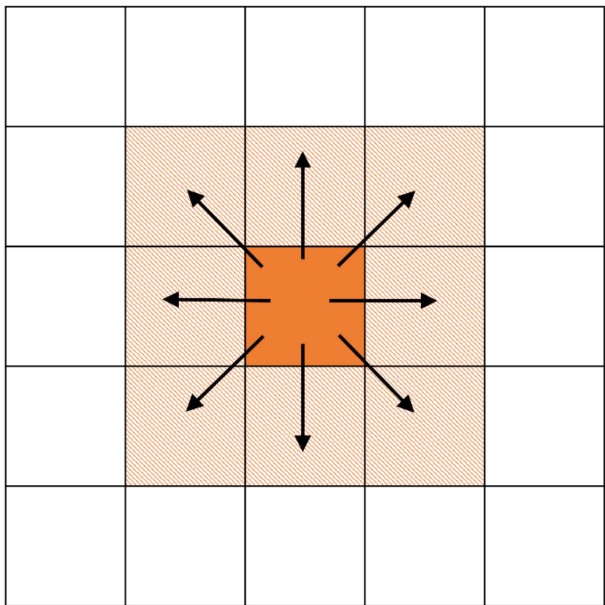

**Figure 1.** First-order queen contiguity relationship between the spatial units.

The numeric-type spatial weight matrix considered in this paper is the numeric threshold distance adjacency matrix. This suggests that the geospatial objects within the specified threshold distance are adjacent, whereas geospatial objects at a distance exceeding the threshold distance are not adjacent. The weight $w_{ij}$ from $s_j$ to $s_i$ is $w_{ij} = \begin{cases} \frac{1}{d_{ij}}, & d_{ij} < d_T \\ 0, & d_{ij} \geq d_T \end{cases}$ , where $d_T$ is the threshold distance.

Considering the influences of all $s_j$ on $s_i$, the spatial weight matrix W should be row standardized. The weighting factors $w_{ij}$ of all $s_j$ on $s_i$ should be standardized to guarantee that their sum equals 1, i.e., $w_{ij} / \left( \sum\limits_{j=1, j \neq i}^{N} w_{ij} \right)$, for $j = 1, 2, \ldots, N$ and $j \neq i$.

### 2.3. Geospatial LS-SVR Model Based on Spatial Autocorrelation

The conventional LS-SVR model can be regarded as a nonlinear regression function:

$$f(x) = \omega^T \varphi(x) + b, \tag{4}$$

where $\omega$ is the weight coefficient vector (column vector), $\varphi(x)$ is the mapping function from the input space to the high-dimensional feature space, and $b$ is the bias term.

The corresponding optimization problem of Equation (4) is:

$$\min_{\omega, b, e} \mathcal{J}(\omega, b, e) = \frac{1}{2} \omega^T \omega + \frac{1}{2} \gamma \sum_{i=1}^{N} e_i^2 = \frac{1}{2} \|\omega\|^2 + \frac{1}{2} \gamma \sum_{i=1}^{N} e_i^2, \text{ subject to } y_i = \omega^T \varphi(x_i) + b + e_i \text{ for } i = 1, 2, \cdots, N, \tag{5}$$

where $\gamma$ is the regularization parameter and $e_i \in R$ is the $i$th error term.

After introducing Lagrange multipliers $\alpha_i$, based on Mercer's condition, we can obtain the final form of the LS-SVR model for nonlinear regression:

$$f(x) = \omega^T \varphi(x) + b = \sum_{i=1}^{N} \alpha_i \langle \varphi(x_i), \varphi(x) \rangle + b = \sum_{i=1}^{N} \alpha_i K(x_i, x) + b, \tag{6}$$

where $K(x_i, x)$ is the kernel function, and $\alpha_i$ and $b$ are regression parameters. The kernel function can be a linear kernel, polynomial kernel, Gaussian kernel, Laplacian kernel, sigmoid kernel, etc.

Similar to the conventional LS-SVR model, the proposed LS-SVR model based on spatial autocorrelation (denoted as the geospatial LS-SVR model or Geo LS-SVR model) can also be regarded as a nonlinear regression function. However, spatial autocorrelation is fused into the regression function $\omega^T \varphi(x) + b$ of the conventional LS-SVR model to ensure that regression modeling of geospatial object $s_i$ considers not only its own explanatory factors $\varphi(x_i)$ but also the explanatory factors $w_{i\varphi} \varphi(x)$ of related objects, where $w_i$ denotes the $i$th row of spatial weight matrix W. The Geo LS-SVR model can be expressed as:

$$f(x) = \omega^T (I + W) \varphi(x) + b, \tag{7}$$

where $(I + W)^T \omega$ is equivalent to the weight vector $\omega$ in Equation (4) of the conventional LS-SVR model [39], I is the identity matrix of N * N.

It should be noted that in SVM theory, the SVR model actually converts a given nonlinear regression problem in the input space into a linear regression problem in the feature space. In terms of the regression problem of geographical data, the input space is the attribute space of geospatial objects $S = \{s_i\}_{i=1}^N$. After the nonlinear regression model $y_i \approx f(x_i, w_{ij}x_j, w_{ij}y_j)$ of the input space is mapped to the linear regression model (Equation (7)) of the feature space via a kernel function, the spatial characteristics, such as the topological structure and spatial location, of geospatial objects in geographic space remain unchanged, i.e., the spatial relationship remains invariant. Therefore, this mapping transformation does not affect the expression of the spatial weight matrix.

Figure 2 shows the relation among the input space, feature space and geographic space.

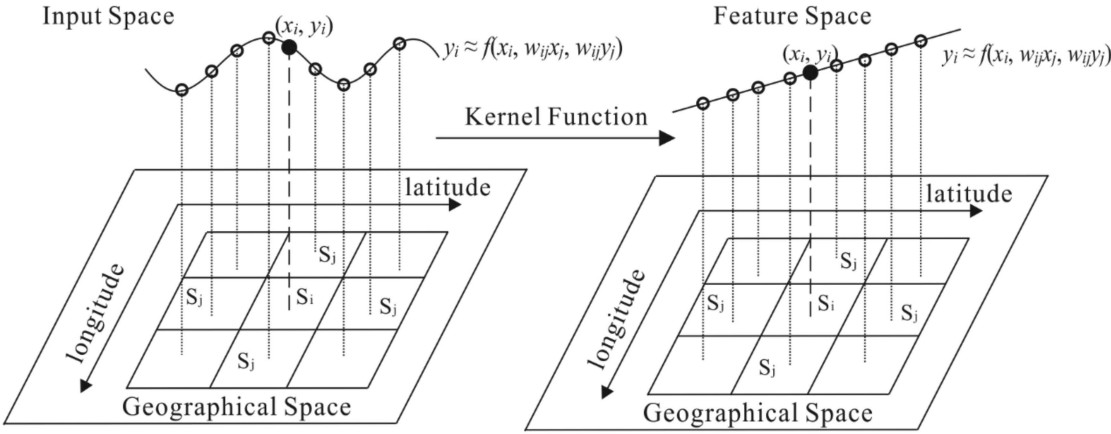

**Figure 2.** Relationship among the input, feature and geographic spaces.

Regarding the Geo LS-SVR model, the corresponding optimization problem of Equation (7) is:

$$\min_{\omega, b, e} \mathcal{J}(\omega, b, e) = \frac{1}{2} \|(I + W)^T \omega\|^2 + \frac{1}{2} \gamma \sum_{i=1}^N e_i^2, \text{subject to } y_i = \omega^T (I + W) \varphi(x_i) + b + e_i \text{ for } i = 1, 2, \cdots, N. \tag{8}$$

After introducing Lagrange multipliers subject to $\alpha_i \in R$, the Lagrangian function of Equation (8) is:

$$\mathcal{L}(\omega, b, e, \alpha) = \mathcal{J}(\omega, b, e) - \sum_{i=1}^N \alpha_i \left( \omega^T (I + W) \varphi(x_i) + b + e_i - y_i \right) = \frac{1}{2} \|(I + W)^T \omega\|^2 + \frac{1}{2} \gamma \sum_{i=1}^N e_i^2 - \sum_{i=1}^N \alpha_i \left( \omega^T (I + W) \varphi(x_i) + b + e_i - y_i \right), \tag{9}$$

where $\alpha = [\alpha_i, \alpha_2, \cdots, \alpha_N]^T$ and I is the identity matrix.

The conditions for optimality are given by:

$$\begin{cases} \frac{\delta\mathcal{L}}{\delta\omega} = 0 \rightarrow (I+W)(I+W)^T\omega - (I+W)\sum_{i=1}^{N}\alpha_i\varphi(x_i) = 0 \\ \frac{\delta\mathcal{L}}{\delta b} = 0 \rightarrow -\sum_{i=1}^{N}\alpha_i = 0 \\ \frac{\delta\mathcal{L}}{\delta e_i} = 0 \rightarrow \alpha_i = \gamma e_i \; for \; i = 1, 2, \cdots, N \\ \frac{\delta\mathcal{L}}{\delta\alpha_i} = 0 \rightarrow \omega^T(I+W)\varphi(x_i) + b + e_i - y_i = 0 \; for \; i = 1, 2, \cdots, N \end{cases} \quad . \tag{10}$$

From the above linear equations, $\omega$ and $e_i$ can be obtained as:

$$\omega = \left((I+W)(I+W)^T\right)^{-1}(I+W)\sum_{i=1}^{N}\alpha_i\varphi(x_i), \tag{11}$$

$$\omega^T = \sum_{i=1}^{N}\alpha_i\varphi(x_i)^T(I+W)^T\left((I+W)(I+W)^T\right)^{-1}. \tag{12}$$

$$e_i = \frac{1}{\gamma}\alpha_i \; for \; i = 1, 2, \cdots, N. \tag{13}$$

After elimination of $\omega$ and $e_i$, the remaining linear equations are:

$$\begin{cases} \sum_{i=1}^{N}\alpha_i = 0 \\ \sum_{i=1}^{N}\alpha_i\varphi(x_i)^T(I+W)^T\left((I+W)(I+W)^T\right)^{-1}(I+W)\varphi(x_i) + b + e_i = y_i \end{cases} \quad . \tag{14}$$

Based on Mercer's condition, the kernel function can be defined as:

$$\Omega_{ij} = \varphi(x_i)^T\varphi(x_j) = K(x_i, x_j) \; for \; i, j = 1, \cdots, N. \tag{15}$$

Let $B = (I+W)^T\left((I+W)(I+W)^T\right)^{-1}(I+W)$, we can define the following:

$$B\Omega = B\cdot\begin{bmatrix} K(x_1, x_1) & K(x_1, x_2) & \cdots & K(x_1, x_N) \\ K(x_2, x_1) & K(x_2, x_2) & \cdots & K(x_2, x_N) \\ \vdots & \vdots & \vdots & \vdots \\ K(x_N, x_1) & K(x_N, x_2) & \cdots & K(x_N, x_N) \end{bmatrix}. \tag{16}$$

Therefore, the above remaining linear equations can be expressed in matrix form as:

$$\begin{bmatrix} 0 & \overline{1}^T \\ \overline{1} & B\Omega\frac{I}{\gamma} \end{bmatrix}\begin{bmatrix} b \\ \alpha \end{bmatrix} = \begin{bmatrix} 0 \\ y \end{bmatrix}, \tag{17}$$

where $\alpha = [\alpha_i, \alpha_2, \cdots, \alpha_N]^T$, $\overline{1} = [1, 1, \cdots, 1]^T$, and $y = [y_1, y_2, \ldots, y_N]^T$.
Let $A = B\Omega + \frac{I}{\gamma}$, and the solutions of the remaining linear equations are:

$$\begin{aligned} b &= \frac{\overline{1}^T A^{-1} y}{\overline{1}^T A^{-1} \overline{1}}, \\ \alpha &= A^{-1}(y - b\overline{1}). \end{aligned} \tag{18}$$

The final form of the Geo LS-SVR model is:

$$f(x) = (I+W)^T\left((I+W)(I+W)^T\right)^{-1}(I+W)\sum_{i=1}^{N}\alpha_i K(x_i, x) + b, \tag{19}$$

where $\alpha_i$ and $b$ are regression parameters.

## 3. Results

### 3.1. Dataset Description and Spatial Dependence Analysis

We adopt three datasets to evaluate the Geo LV-SVR model, including a Boston housing dataset [40], real estate transactions dataset [41] and election dataset [42].

The Boston housing dataset includes house price data derived from the U.S. Census Service concerning housing in the area of Boston, MA, in the mid-1970s, which is often considered to conduct experiments involving machine learning methods on regression problems. Thirteen attributes, such as the average number of rooms per dwelling, crime rate and index of accessibility to radial highways, are considered to predict the median values of owner-occupied homes (in USD 1000) along N = 506 census tracts in Boston.

The real estate transactions dataset contains information on 985 sales in the Greater Sacramento area over a period of five consecutive days. The house price is related to the number of bedrooms and bathrooms, floor space and geographical location of the house. However, as often occurs with missing data problems in real data applications, 18% of home sales exhibit at least one missing feature. To ensure the model training accuracy, we only retain data without any missing features, for a total value of N = 806.

In the election dataset, the population ratios of homeownership, income and individuals with college degrees in N = 3107 counties are considered to predict the vote casting rate during the 1980 USA presidential election.

The Boston housing and real estate transactions datasets are both point feature datasets, and the spatial adjacency matrix cannot be calculated. Therefore, we generate Tyson polygons from point features to convert these features into polygon features. The data distribution of the three datasets is shown in Figure 3. The dependent and explanatory variables involved in these three datasets are listed in Table 1.

Spatial autocorrelation analysis of the abovementioned three datasets should be performed before we apply the Geo LS-SVR model to ensure spatial independence of the geospatial objects. Global Moran's I can measure spatial autocorrelation based on both location data and feature values as follows:

$$I = \frac{N \sum_{i=1}^{N} \sum_{j=1}^{N} w_{ij}(y_i - \overline{y})(y_j - \overline{y})}{\sum_{i=1}^{N} \sum_{j=1}^{N} w_{ij} \sum_{i=1}^{N}(y_i - \overline{y})^2}, \tag{20}$$

where $N$ is the number of geospatial objects, $y_i$ is the value of attribute $y$ for the $i$-th object, $\overline{y}$ is the mean of the attribute value of $y$, and $w_{ij}$ denotes the $i$-th row and $j$-th column element of spatial weight matrix $W$. The value range of Moran's I index is $(-1,1)$. A Moran's I value greater than 0 indicates a positive spatial autocorrelation among the geospatial objects. In contrast, a Moran's I value less than 0 indicates a negative autocorrelation. The greater the absolute value is, the stronger the spatial autocorrelation.

A significance level test is required after calculation of Moran's I values. Generally, the standardized z statistic is considered to assess the reliability of the conclusion that there exists spatial autocorrelation between geospatial objects [43]. The *p* value of the hypothesis test is calculated and compared to the significance level $\alpha$. Usually, we assume a significance level of $\alpha = 0.05$, which suggests that there occurs a significant spatial autocorrelation if $|z| \geq 1.96$. Positive and significant z-values indicate that there exists a positive spatial autocorrelation, and geospatial objects tend to be clustered. A negative and significant z-value indicates that there occurs a negative spatial autocorrelation and that geospatial objects tend to be dispersed. When the z-value is zero, the distribution is random and independent.

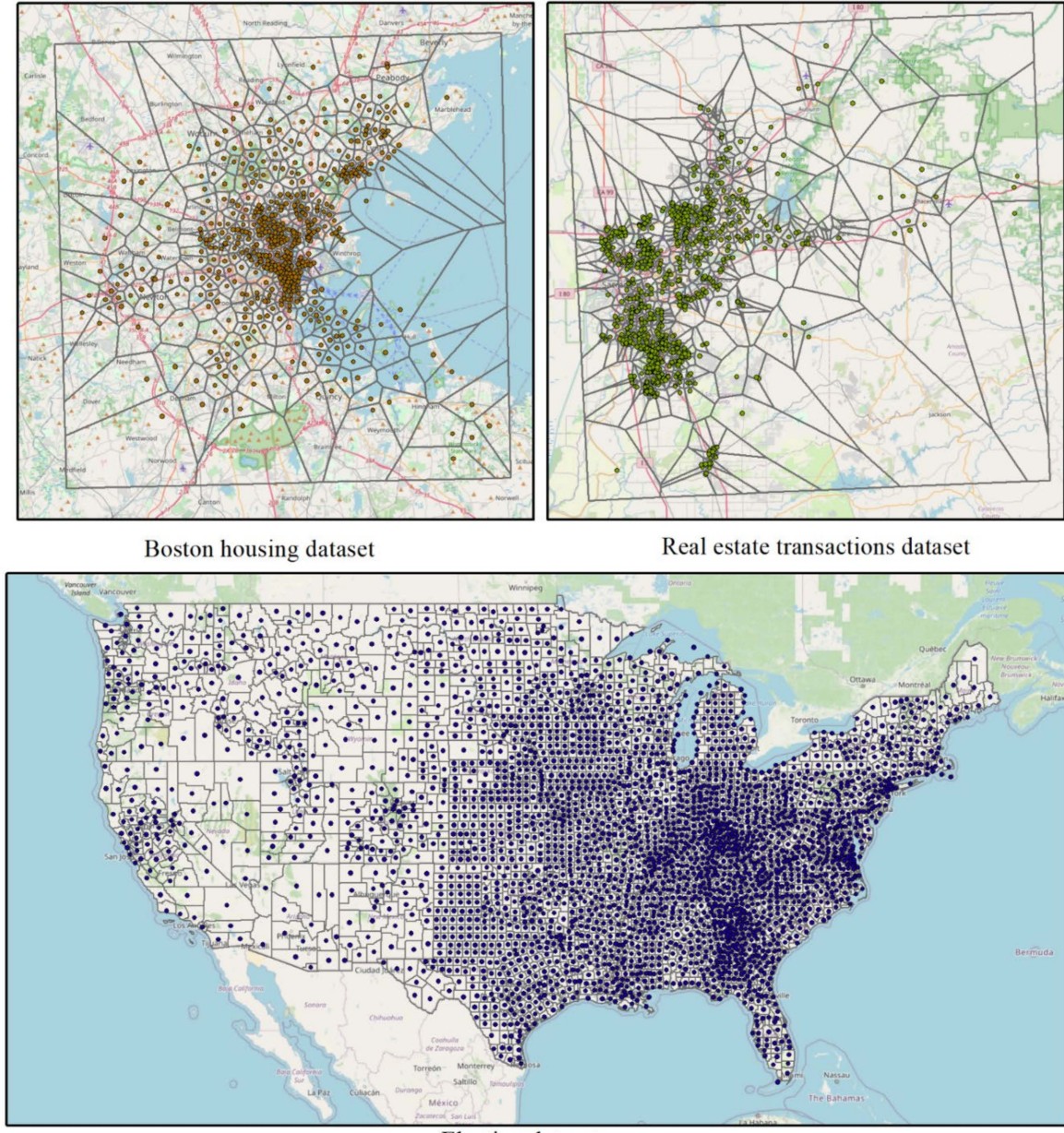

**Figure 3.** Data distribution of the Boston housing, real estate transactions and election datasets.

Global Moran's I values for the above three datasets with a first-order queen adjacency matrix were calculated: global Moran's I for dependent variable medv in the Boston housing dataset was 0.584. The z-score calculated based on the random null hypothesis was 21.892. Similarly, regarding dependent variable price in the real estate transactions dataset, Global Moran's I reached 0.551, and z reached 28.805. In terms of dependent variable vote casting ratio in the election dataset, the following values were obtained: Global Moran's I = 0.608, and z = 58.086. The *p* values of these three datasets were all less than 0.0001. These indicators suggested that the spatial distribution of the dependent variables of these three datasets exhibits a significant clustering pattern, and the probability of randomly generating this clustering pattern is lower than 1%.

**Table 1.** Variable information on the three datasets.

| | Boston Housing Dataset (N = 506) | | Real Estate Transactions Dataset (N = 806) | | Election Dataset (N = 3107) | |
|---|---|---|---|---|---|---|
| Variable Type | Variable Name | Variable Meaning | Variable Name | Variable Meaning | Variable Name | Variable Meaning |
| Dependent variable | medv | Median value of owner-occupied homes in USD 1000 | price | Home sales price in USD 10,000 | Vote casting ratio | Population-wide cast votes/population over age 19 eligible to vote |
| Explanatory variable | Crim | Per capita crime rate by town | beds | Number of bedrooms | College degree ratio | Population with college degrees/population over age 19 eligible to vote |
| | Zn | Proportion of residential land zoned for lots larger than 25,000 sq. ft | | | | |
| | indus | Proportion of nonretail business acreage per town | | | | |
| | Chas | Charles River dummy variable (=1 if the tract bounds the river; 0 otherwise) | | | | |
| | Nox | Nitric oxide concentration (parts per 10 million) | | | | |
| | Rm | Average number of rooms per dwelling | | | | |
| | Age | Proportion of owner-occupied units built prior to 1940 | baths | Number of bathrooms | Homeownership ratio | Homeownership/population over age 19 eligible to vote |
| | Dis | Weighted distances to five Boston employment centers | | | | |
| | Rad | Index of accessibility to radial highways | | | | |
| | Tax | Full-value property-tax rate per USD 10,000 | | | | |
| | Ptratio | Pupil-teacher ratio by town | sq_ft | Square footage | Per capita income | Income/population over age 19 eligible to vote |
| | B | 1000 (Bk-0.63)^2, where Bk is the proportion of African Americans by town | | | | |
| | Lstat | Percentage of the population with a lower status | | | | |

In addition, we also calculated local Moran's I indicator values (Equation (21)) [44] for the dependent variables of the three datasets, and a scatter diagram is shown in Figure 4. The scatter points in the three figures are concentrated in the first and third quadrants, which indicates that high–high and low–low clustering patterns are common distributions for most of the geospatial objects. For a geospatial object with a high dependent variable value, the surrounding geospatial objects also exhibit a high dependent variable value. Conversely, for a geospatial object with a low dependent variable value, the surrounding geospatial objects also exhibit a low dependent variable value. There occur significant spatial dependencies between the geospatial objects and surrounding geospatial objects. Information regarding the surrounding objects is beneficial to predict dependent variable values of the geospatial objects. Therefore, it could be considered that the Geo LS-SVR model fused with the spatial weight matrix may be more suitable for nonlinear analysis of these three datasets.

$$I_i = \frac{N^2(y_i - \bar{y})\sum_{j\neq i}^n w_{ij}(y_i - \bar{y})}{\sum_{i=1}^N \sum_{j=1}^N w_{ij}\sum(y_i - \bar{y})^2} \tag{21}$$

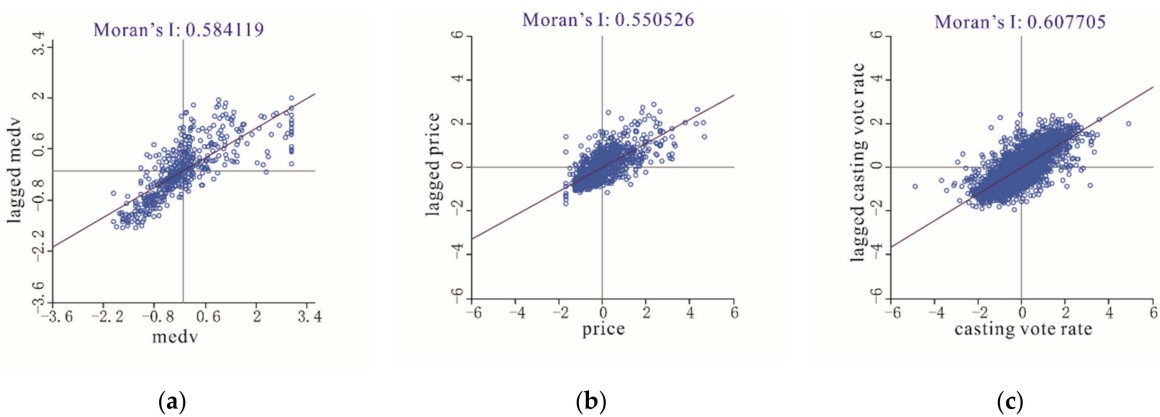

|       (a)       |       (b)       |       (c)       |

**Figure 4.** Moran's I scatter diagram of the dependent variables of these three datasets: (**a**) Boston housing dataset; (**b**) real estate transactions dataset; (**c**) election dataset.

### 3.2. Experimental Results and Evaluation

In this section, we conducted experiments on the two spatial weight construction methods of the Geo LS-SVR model and compared them to the traditional LS-SVR model. In addition, the numeric-type Geo LS-SVR model was compared to other common models to evaluate the effect of the proposed model.

#### 3.2.1. Model Evaluation Index

The mean absolute error (MAE), root mean square error (RMSE) and coefficient of determination ($R^2$) were adopted in this experiment to evaluate the prediction performance. Regarding the spatial regression problem, the true value is $y = \{y_1, y_2, \cdots, y_N\}$ and the predicted value is $\hat{y} = \{\hat{y}_1, \hat{y}_2, \cdots, \hat{y}_N\}$. Hence, the above metrics can be calculated as:

$$\text{MAE} = \frac{1}{N}\sum_{i=1}^N |y_i - \hat{y}_i|, \tag{22}$$

$$\text{RMSE} = \sqrt{\frac{1}{N}\sum_{i=1}^N (y_i - \hat{y}_i)^2}, \tag{23}$$

$$R^2 = 1 - \frac{\sum_{i=1}^N (y_i - \hat{y}_i)^2}{\sum_{i=1}^N (y_i - \bar{y}_i)^2}. \tag{24}$$

Both MAE and RMSE can reflect the error between the real and predicted values, where MAE reflects the true error in the whole test set and RMSE is more affected by outliers. The smaller the MAE and RMSE values are, the higher the prediction accuracy. $R^2$ reflects how well the model fits the true values. $R^2$ indicates the explanatory ability of different models, with a value range of $[-\infty, 1]$. $R^2 = 0$ suggests that the model provides the same effect as does a model only calculating the average explanatory variable value as the predicted value. $R^2 = 1$ indicates that the model perfectly fits the true values, and each predicted value is the same as the true value. The closer $R^2$ is to 1, the better the model fitting effect to the data. Its dimensions remain unchanged with changes in the datasets and models and can be used to evaluate the performance of different models considering various datasets.

### 3.2.2. 0–1 Type Fusion Process and Result Analysis

In contrast to the conventional LS-SVR model, the Geo LS-SVR model with 0–1 type fusion (denoted as the 0–1 type Geo LS-SVR model) must integrate the spatial adjacency relationships among the geospatial objects. To maintain invariance of the adjacency relationships when randomly dividing the training/test and sampling sets, spatial K-fold cross validation [45] was adopted. K subsamples with an equal number of geospatial objects were roughly divided from the dataset. K training and testing operations were carried out. At each time point, one subsample was used for prediction, and the remaining K-1 subsamples were used for training. On this basis, the data points of training set which is close to the test set are removed to ensure that the training dataset only contains data points that are far away from the test dataset. Generally, K is set to 5 to obtain the proper ratio of training to test samples. For example, five subsamples were divided from the election dataset (Figure 5), and the division method for the other two datasets was similar.

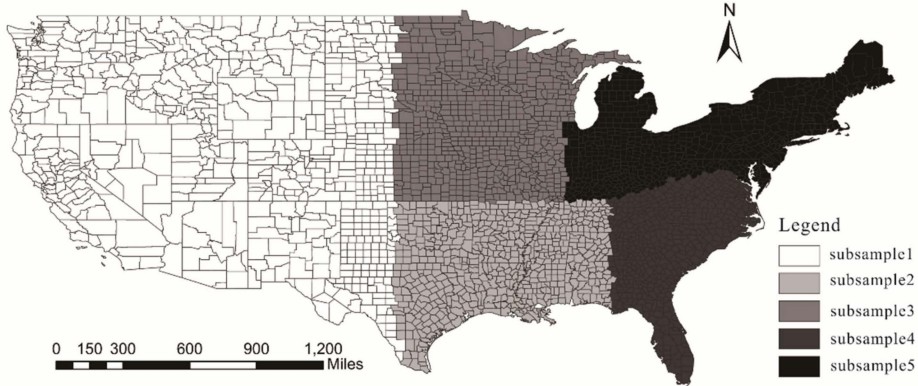

**Figure 5.** Subsample distribution diagram of the election dataset.

In this experiment, the first-order queen adjacency matrix mentioned in Section 3.2.1 was selected as the spatial weight matrix fusion method, which is a 0–1 type fusion method. To ensure fairness of the evaluation experiment, a Gaussian kernel was adopted as the kernel function of the SVR model in the training process of these four models.

The prediction results for the Boston housing, real estate transactions and election datasets are shown in Figures 6–8, respectively. To intuitively reveal the effects of these models, the prediction results were arranged according to their true values in descending order. Model evaluation index values for these three datasets were calculated (Table 2).

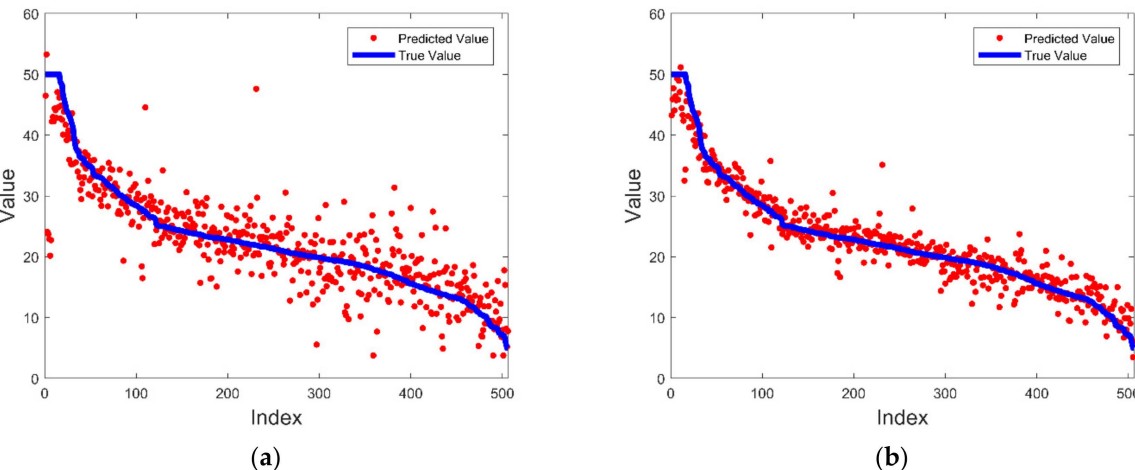

**Figure 6.** Boston housing dataset prediction results obtained with the models fused with the 0–1 type spatial weight matrix: (**a**) conventional LS-SVR model; (**b**) Geo LS-SVR model.

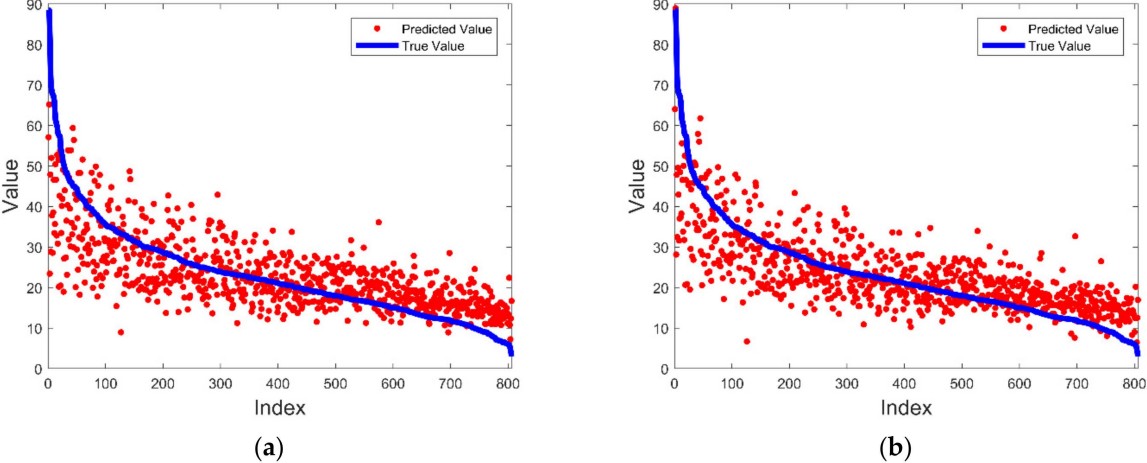

**Figure 7.** Real estate transactions dataset prediction results obtained with the models fused with the 0–1 type spatial weight matrix: (**a**) conventional LS-SVR model; (**b**) Geo LS-SVR model.

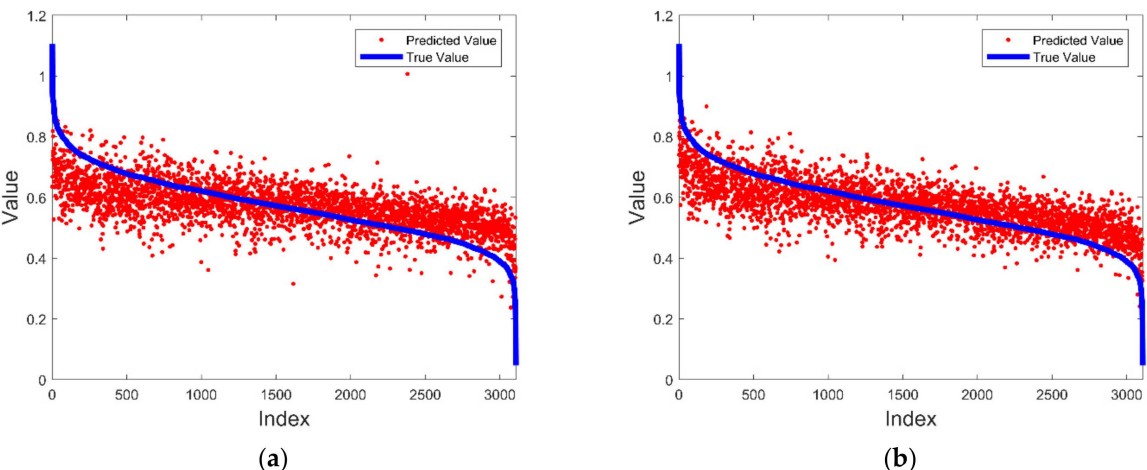

**Figure 8.** Election dataset prediction results obtained with the models fused with the 0–1 type spatial weight matrix: (**a**) conventional LS-SVR model; (**b**) Geo LS-SVR model.

**Table 2.** Evaluation index value comparison between the conventional LS-SVR and 0–1 type Geo LS-SVR models.

| Evaluation Index | Model | Boston Housing Dataset | Real Estate Transactions Dataset | Election Dataset |
|---|---|---|---|---|
| MAE | Conventional LS-SVR | 3.2196 | 6.1166 | 0.0653 |
| | Geo LS-SVR | 1.7617 | 5.7292 | 0.0576 |
| RMSE | Conventional LS-SVR | 4.9076 | 8.4198 | 0.0837 |
| | Geo LS-SVR | 2.5676 | 7.9413 | 0.0735 |
| $R^2$ | Conventional LS-SVR | 0.7138 | 0.5220 | 0.3982 |
| | Geo LS-SVR | 0.9216 | 0.5748 | 0.5383 |

The results of these three indicators reveal that the performance of the Geo LS-SVR model is superior to that of the conventional LS-SVR model for all three datasets. Comparing these three datasets, the performance of the Geo LS-SVR model on the Boston housing and election datasets is greatly improved. The MAE index value is reduced by 45.28% and 55.44%, respectively, the RMSE index value is reduced by 47.68% and 56.75%, respectively, and the $R^2$ index value is increased by 29.11% and 35.11%, respectively. However, the real estate transactions dataset is difficult to predict because the real estate transactional data are not processed, and there is a large difference between the values of the variables. Neither the conventional LS-SVR model nor the Geo LS-SVR model performs well. The RMSE index value is approximately 8 (i.e., the average gap between the predicted and real values of the housing price is approximately USD 8000), and the value is small for higher prices but large for lower prices.

### 3.2.3. Numerical Distance Type Fusion Process and Result Analysis

The application of the numeric threshold distance adjacency matrix mentioned in Section 3.2.2 as the numeric-type spatial weight matrix requires us to determine an appropriate threshold distance to reflect the real changes in spatial phenomena before the experiments. Incremental spatial autocorrelation analysis refers to the calculation of the global spatial autocorrelation for a series of increasing distances and measurement of the intensity of spatial clustering for each distance based on the z-score returned [46]. The z-score generally peaks, which reflects those distances where the spatial processes promoting clustering are the most pronounced. Therefore, the threshold distance should be set to the maximum peak distance. Choosing the Boston housing dataset as an example, it can be found from the incremental spatial autocorrelation line chart (Figure 9) that with increasing distance, the z-score exhibits two peaks, and the maximum peak distance is 7827 m. Therefore, the threshold distance used in the experiment on the Boston housing dataset is set to 7827 m. The threshold distances for the other two datasets are determined in a similar way to the above approach.

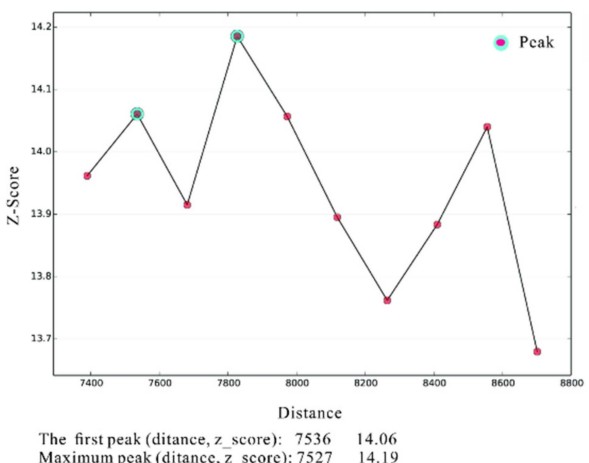

The first peak (ditance, z_score): 7536   14.06
Maximum peak (ditance, z_score): 7527   14.19

**Figure 9.** Incremental spatial autocorrelation line chart for the Boston housing dataset.

To ensure the allocation rationality of the training and test sets, there are 405 units in the training set and 101 units in the test set for the Boston housing dataset, 624 units in the training set and 182 units in the test set for the real estate transactions dataset, and 2485 units in the training set and 622 units in the test set for the election dataset. All of the training and test sets were randomly selected, and the arranged prediction results obtained with the conventional LS-SVR model and the Geo LS-SVR model fused with the numeric-type spatial weight matrix (denoted as the numeric-type Geo LS-SVR model) are shown in Figures 10–12. Model evaluation index values of these models were calculated for the above three datasets (Table 3).

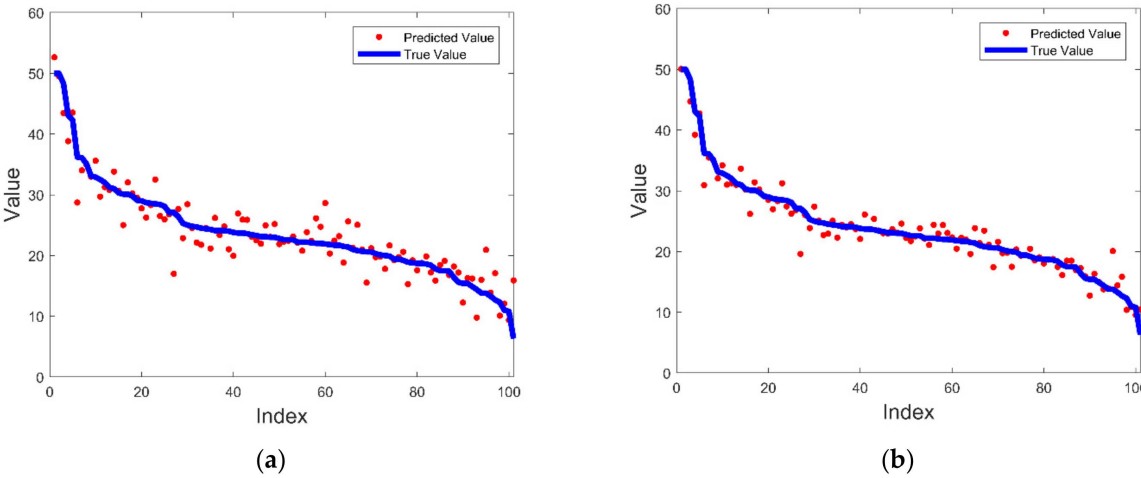

(**a**)                                                        (**b**)

**Figure 10.** Boston housing dataset prediction results obtained with the models fused with the numeric-type spatial weight matrix: (**a**) conventional LS-SVR model; (**b**) Geo LS-SVR model.

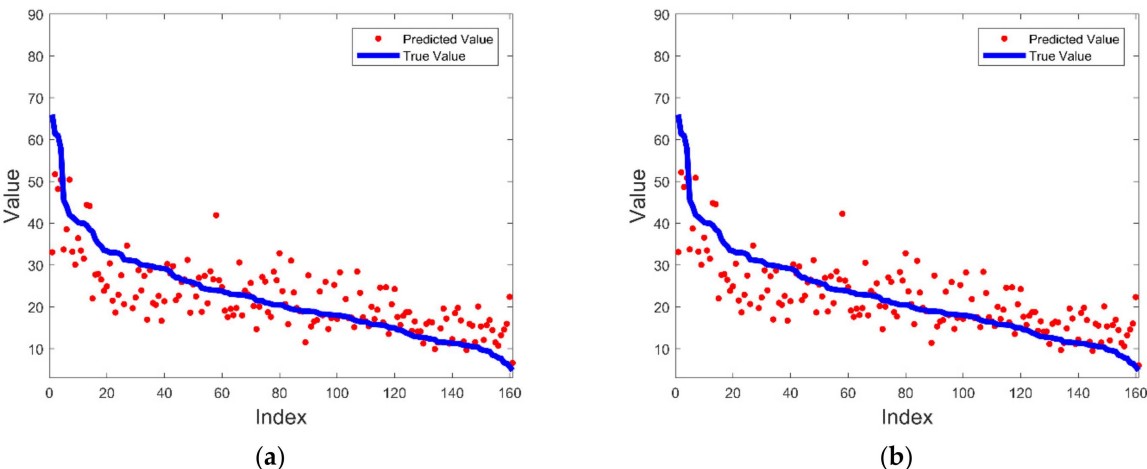

(**a**)                                                        (**b**)

**Figure 11.** Real estate transaction dataset prediction results obtained with the models fused with the numeric-type spatial weight matrix: (**a**) conventional LS-SVR model; (**b**) Geo LS-SVR model.

Overall, the performance of the Geo LS-SVR model is superior to that of the conventional LS-SVR model. The dot graph of the prediction results for these datasets shows that the prediction points determined with the Geo LS-SVR model are more clustered around the real value curve, which suggests that the fitness of the Geo LS-SVR model is better than that of the other models. Similar to the results in Section 3.2.1, the Geo LS-SVR model achieves a great improvement on the Boston housing and election datasets. The RMSE index values of the conventional LS-SVR and Geo LS-SVR models are always higher than the MAE index values, but the RMSE index values of the Geo LS-SVR model are

closer to the MAE index values, indicating that the Geo LS-SVR model is more sensitive to large errors.

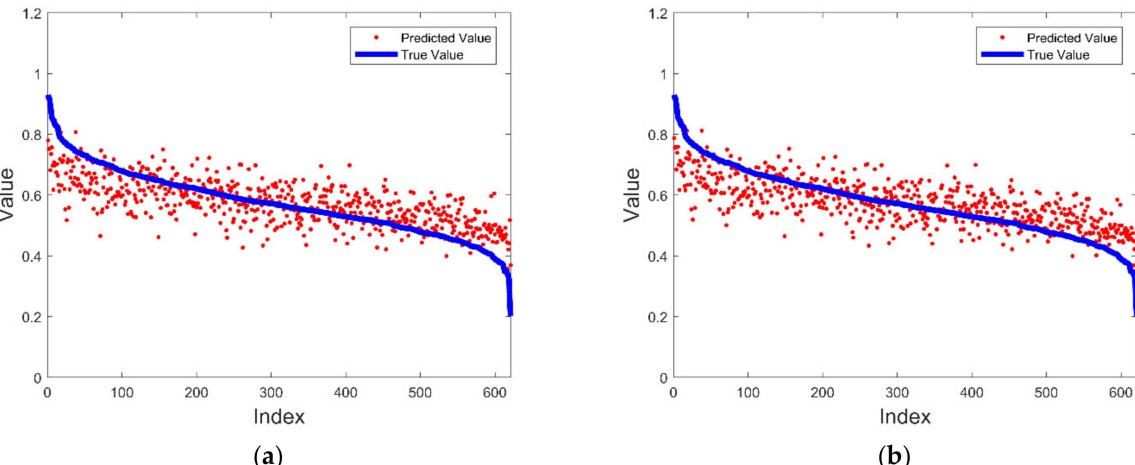

(**a**)  (**b**)

**Figure 12.** Election dataset prediction results obtained with the models fused with the numeric-type spatial weight matrix: (**a**) conventional LS-SVR model; (**b**) Geo LS-SVR model.

**Table 3.** Evaluation index value comparison between the conventional LS-SVR and numeric-type Geo LS-SVR models.

| Evaluation Index | Model | Boston Housing Dataset | Real Estate Transactions Dataset | Election Dataset |
|---|---|---|---|---|
| MAE | LS-SVR | 2.0317 | 5.2165 | 0.0595 |
| | Geo LS-SVR | 1.2250 | 5.2072 | 0.0587 |
| RMSE | LS-SVR | 2.8182 | 6.8265 | 0.0748 |
| | Geo LS-SVR | 1.8205 | 6.8179 | 0.0735 |
| $R^2$ | LS-SVR | 0.8676 | 0.6091 | 0.5178 |
| | Geo LS-SVR | 0.9448 | 0.6101 | 0.5351 |

### 3.2.4. Comparison to Traditional Models and Result Analysis

The Boston housing dataset has been extensively described from many statistical perspectives [47,48]. To more accurately evaluate the prediction performance of the Geo LS-SVR model, we compared the results obtained with the numeric-type Geo LS-SVR model to those obtained with models widely applied in regression prediction problems on the Boston dataset, as follows:

1. Linear regression: We calculated the correlation coefficients between 13 independent variables and the dependent variable, chose three variables with absolute values of the correlation coefficient with dependent variable medv greater than 0.5, i.e., RM, PTRATIO and LSTAT, and employed these three variables to perform linear regression.
2. Ridge regression: Multivariate polynomial fitting with a degree of 3 was applied, and ridge regression was adopted as the regression method.
3. Decision tree regression [49]: A tree structure was used for regression. The decision process of the decision tree starts from the root node of the tree, the data were compared to feature nodes, and the next comparison branch was selected according to the comparison result until the leaf node providing the final decision result was obtained. RMSE was chosen as the criterion. The maximum depth was set to 9.
4. K-nearest neighbor regression (KNN) [50]: A nonparametric model that only makes decisions on the regression values of the test samples with the help of the target values of K nearest training samples. We set the number of neighbors to 5 and the weight to the inverse distance weight.

5.  SVR: In the conventional SVR model, a Gaussian kernel was selected as the kernel function.

The prediction performance of each model is summarized in Table 4.

**Table 4.** Evaluation index value comparison between the numeric-type Geo LS-SVR model and traditional models.

| Model | MAE | RMSE | $R^2$ |
|---|---|---|---|
| Linear regression | 3.7668 | 5.4849 | 0.6550 |
| Ridge regression | 2.7778 | 4.3512 | 0.7829 |
| Decision tree regression | 1.2753 | 2.4832 | 0.9288 |
| KNN regression | 0.77034 | 2.3486 | 0.9367 |
| SVR | 2.4802 | 3.1604 | 0.8846 |
| LS-SVR | 2.0317 | 2.8182 | 0.8676 |
| Geo LS-SVR | 1.2250 | 1.8205 | 0.9448 |

The independent and dependent variables of the Boston housing dataset attain not only linear relationships but also nonlinear relationships. Therefore, in terms of the linear and ridge regression models, which can only capture linear relationships, the model prediction performance is poor. In contrast, the MAE and RMSE index values of the remaining five models are all lower, and the $R^2$ index values are all above 0.8, indicating a good prediction performance of these models. The MAE index value of the K-nearest neighbor regression model is lower, but the RMSE index is higher, indicating that its overall error is small, but the model is insensitive to certain large errors. Correspondingly, the MAE index value of the Geo LS-SVR model is higher than that of the K-nearest neighbor regression model, but the RMSE index value is the smallest and the $R^2$ index value is the highest among these models. This suggests that the Geo LS-SVR model can better fit the dataset, and the proposed model performs better in dependent variable prediction.

## 4. Discussion

Comparing the $R^2$ index values between the 0–1 type and numeric-type Geo LS-SVR models, it is found that the numeric-type fusion method yields a good fitting effect. This may occur because 0–1 type spatial weight matrix is binarized and only contains the information of first order neighbors. However, the numeric-type Geo LS-SVR model considers more surrounding geospatial objects at locations where the geospatial objects are clustered. The weight of numeric-type spatial weight matrix will decrease with distance increasing. It is consistent with Tobler's First Law of Geography that closer objects are more strongly related than are more distant ones.

Due to the complex data structure of spatial lattice data, there are nonlinear relationships among variables. Models which can only capture linear relations have a poor fitting effect on data with nonlinear relations. SVR, LS-SVR and Geo LS-SVR models use kernel function to map input data to high-dimensional space and perform linear regression in high-dimensional space which facilitates mathematical calculations. From the results in Section 3.2.4, we can find that the ability of learning nonlinear relationships makes SVR, LS-SVR and Geo LS-SVR models perform better on datasets with nonlinear relationships among variables.

Combined with the experimental results in Sections 3.2.2 and 3.2.3, the Geo LS-SVR model always achieves a better prediction result than does the conventional model regardless of the dataset or type of spatial weight matrix. The conventional LS-SVR model only performs nonlinear relationship mapping of an object without considering other spatially adjacent or nearby objects. In the comparison of nonlinear models, the Geo LS-SVR model with spatial weight matrix is sensitive to some large errors and can better fit spatial datasets. The KNN model uses neighbor samples for prediction, but the neighbor refers to objects which have higher numerical similarity of independent variables rather than adjacent or nearer in space. Spatial weight matrices represent the spatial autocorrelation between

dependent variables of geospatial objects. The fusion with spatial weight matrices enables the Geo LS-SVR model to predict based on the information of the geospatial object to be predicted and its neighbor geospatial objects. The ability of geospatial data to describe geospatial phenomena is also improved. We can infer that considering spatial autocorrelation of spatial lattice data when implementing the LS-SVR model is feasible and useful. The Geo LS-SVR model can solve practical problems by comprehensively considering the spatial and attribute characteristics of geospatial geographic objects. It is more suitable for regression prediction of spatially dependent geospatial objects.

In future research, we may fuse the spatial weight matrix via other fusion methods or modify the calculation form of the spatial weight matrix. In addition, since spatial autocorrelation among geospatial objects can be reflected in the model, further research on whether other spatial information, such as spatial heterogeneity, sequential relationships, metric relationships and topological relationships, can be reflected in the model will constitute a greater challenge.

**Author Contributions:** Conceptualization, Haiqi Wang and Lei Che; methodology, Haiqi Wang, Liuke Li and Lei Che; model code writing, Liuke Li and Lei Che; data curation, Haoran Kong, Qiong Wang, Zhihai Wang and Jianbo Xu; validation, Liuke Li and Lei Che; writing—original draft preparation, Lei Che; writing—review and editing, Haiqi Wang and Liuke Li; visualization, Liuke Li and Haoran Kong; supervision, Haiqi Wang; funding acquisition, Haiqi Wang and Lei Che. All authors have read and agreed to the published version of the manuscript.

**Funding:** This research was funded by the National Natural Science Foundation of China (grant number 41471322) and the National Key R&D Program of China (grant number 2018YFB0505100).

**Institutional Review Board Statement:** Not applicable.

**Informed Consent Statement:** Not applicable.

**Data Availability Statement:** Publicly available datasets were analyzed in this study. These datasets can be accessed as follows: a. Boston housing dataset (https://archive.ics.uci.edu/mL/machine-learning-databases/housing accessed date: 27 September 2021); b. real estate transactions dataset (https://support.spatialkey.com/wp-content/uploads/2021/02/Sacramentorealestatetransactions.csv accessed date: 27 September 2021); c. election dataset (http://www.spatial-econometrics.com/data/contents.html accessed date: 27 September 2021).

**Acknowledgments:** We really appreciate the editor and two anonymous reviewers for their useful comments and suggestions. 

**Conflicts of Interest:** The authors declare no conflict of interest.

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
