# Peer review of "Geospatial Least Squares Support Vector Regression Fused with Spatial Weight Matrix"

_ijgi, doi:10.3390/ijgi10110714_

Round 1
Reviewer 1 Report
Minor text editing is needed.
Author Response
Thank you for your comment. As suggested by the reviewer, we have reviewed the manuscript and did a text editing.
Reviewer 2 Report
The authors propose a geospatial prediction model based on least squares support vector regression (LS-SVR), to support regression predictions of spatially dependent geospatial objects. The model was tested on three datasets and compared to the conventional LS-SVR model, with the conclusion that the developed geospatial LS-SVR model performs a better prediction for spatially dependent objects. The paper is well structured, well written and clearly presents its methods and results.
Author Response
Thank you for your comments, we are very glad to get your approval.
Reviewer 3 Report
Review of the manuscript “Geospatial least squares support vector regression fused with spatial weight matrix”
This study aimed to propose a geospatial LS-SVR model for geospatial data regression prediction. Generally, the structure of the text is correct and the topic of the article is interesting.
Methods should be described more clearly. What are assumptions of this model?
Discussion section is missing. In Discussion section authors should refer to the literature mentioned in the Introduction section.
Please carefully check the language when you prepare to submit a revised version.
The article fits the scope of International Journal of Geo-Information. The topic of the article is interesting and up-to-date.
Author Response
Thank you for your comments. We have revised Section 2.1 Problem Description in the new manuscript. We hope that model assumptions have been described more clearly. We also added a discussion section to discuss the model at a deeper level. The whole manuscript has been revised to avoid language mistakes.
This manuscript is a resubmission of an earlier submission. The following is a list of the peer review reports and author responses from that submission.
Round 1
Reviewer 1 Report
Review of
Geospatial least squares support vector regression based on spatial correlation
General comment
This paper study LS-SVR based on data spatially correlated. The paper is organized into 6 sections: First an introduction about machine learning for spatial data. Section 2 shows a review of related works. In Section 3 is given a short review about LS-SVR. In Sections 4 and 5 are presented both the methodology proposed and applications to real data. The paper ends with a conclusion section. The manuscript covers an interesting topic for IJGI. There are some minor points, which must be treated. I recommend minor revisions.
Specific comments
- A more specific revision in Section 2 must be given.
 - The methodology presented is useful for Areal Data (this point must be clarified). Geostatistics and point patterns are other fields of spatial statistics which are not considered.
 - Line 153: Spatial correlation between any two geographic objects ?? and ?? can be quantitatively measured by the spatial weight matrix WN×N. This is only true for areal data
 - Lines 167 to 222. All the review is valid only for areal data. The weights matrices are defined in the context of areal data. In this case the spatial dependence is defined in terms of contiguity or distance between centroids. A discussion in this point is required.
 - Equation 7 is valid only in the context of areal data
 - Review notation in Equation 8
 - In Equations (9) , (10), and (11) there are some terms which are not previously defined.
 - Boston housing dataset has been widely described using from many statistical (parametric and non-parametric) perspectives. It would be useful to highlight the advantages of the methodology proposed. A comparison with classical
 - Other criteria in addition to MSE can be considered to evaluate the performance?

Reviewer 2 Report
I enjoyed reading the paper: Geospatial least squares support vector regression based on spatial correlation. It claims that the proposed model is suitable for geographical data regression prediction and improves the geospatial phenomena explanation ability of geospatial data. I, nonetheless, have some concerns about this paper; they are as follows:
- Seemingly, authors are confused between spatial correlation and spatial autocorrelation while both concepts are rich in the literature. See:

Legendre, Pierre. "Spatial autocorrelation: trouble or new paradigm?." Ecology 74.6 (1993): 1659-1673.
Anselin, Luc. "Local indicators of spatial association—LISA." Geographical analysis 27.2 (1995): 93-115.
- The structure of the paper is arbitrary. For example, where is Methodology? Restructuring and summarizing the paper are highly recommended.
 - There is no balance between different sections of the paper. All sections of the paper are poorly written, especially Introduction and Conclusion. Problem definition must be explained clearly in the Introduction section. Generally speaking, many parts of the text are suffering from a lack of coherence. Some sentences are too long and hard to follow. The text of this paper, in general, needs a thorough review, as there are multiple typos and grammatical errors
 - Abbreviations must be identified first for the reader when they appear in the text for the first time, e.g., WSVM, etc.
 - Literature Review must be improved. There is no coherence between different studies. The authors cited some research randomly without a clear aim.
 - Authors should cite original references. Considering that this is a technical paper, and the authors discussed different concepts and techniques, 37 references are not enough.
 - The methodology and the innovation of the paper are confusing and ambiguous. Each method or technique should be clearly and briefly explained. All formulas must be explained in a clear way for the reader and cited by original references. There is too much emphasis on unnecessary details in some sections of the paper as well.
 - The authors have conclusions based on unclear connections with the results and supporting assumptions and theories.
9. The abstract needs rewording and revision, especially the definition of the First Law of Geography 
Authors: According to the Tobler's First Law of Geography, all attribute values on a geographic surface are related to each other.
While the original definition is:
"everything is related to everything else, but near things are more related than distant things
Tobler, Waldo R. "A computer movie simulating urban growth in the Detroit region." Economic geography 46.sup1 (1970): 234-240.
- The case study and is not clear. The authors should present the case study very well. Where is the map of the case study? Date of data? See below:

5.1 Datasets description
5.1.1 Three datasets
We use three datasets to evaluate Geo LV-SVR, including Boston housing dataset [34], real estate transactions dataset [35] and elect dataset [36].
- The size of Figure 1 is large. The description of Figure 3 is very short.

Based on the above comments, I think this paper cannot be published in its current form.

Reviewer 3 Report
The authors proposed the geospatial Least Squares Support Vector Regression (LS-SVR) model which is a LS-SVR model that integrates spatial correlations of geospatial objects to make the regression prediction of geospatial data. Two different types of spatial weight matrixes were used with the regression function of LS-SVR to measure the strength of interaction between spatial objects.
The authors used well-known machine learning datasets to evaluate the accuracy of the developed Geospatial LS-SVR model and compare it to the conventional LS-SVR model. The results show that using spatial correlation could improve the prediction accuracy compared to ordinary LS-SVR. This is an interesting idea, and I suggest wider discussion of the usability of this approach in the real-world scenarios which will support the strength of the approach.
There are some typos, grammatical errors and blank spaces, so I suggest additionally checking the text.